

# Future summer warming pattern under climate change is regulated by lapse-rate changes

Roman Brogli[1], Silje Lund Sørland[1,2], Nico Kröner[1,3], and Christoph Schär[1]

[1]Institute for Atmospheric and Climate Science, ETH Zurich, Universitätstrasse 16, 8092 Zurich, Switzerland
[2]NORCE, Jahnebakken 5, 5007 Bergen, Norway
[3]South Pole, Technoparkstrasse 1, 8005 Zurich, Switzerland

**Correspondence:** Roman Brogli (roman.brogli@env.ethz.ch)

**Abstract.** Greenhouse gas-driven global temperature change projections exhibit spatial variations, meaning that certain land areas will experience substantially enhanced or reduced surface warming. It is vital to understand enhanced regional warming anomalies as they locally increase heat-related risks to human health and ecosystems. We argue that tropospheric lapse-rate changes play a key role in shaping the future summer warming pattern around the globe in mid-latitudes and the tropics.
We present multiple lines of evidence supporting this finding based on idealized simulations over Europe, as well as regional and global climate model ensembles. All simulations consistently show that the vertical distribution of tropospheric summer warming is different in regions characterized by enhanced or reduced surface warming. Enhanced warming is projected where lapse-rate changes are small, implying that the surface and the upper troposphere experience similar warming. On the other hand, strong lapse-rate changes cause a concentration of warming in the upper troposphere and reduced warming near the surface. The varying magnitude of lapse-rate changes is governed by the temperature dependence of the moist-adiabatic lapse rate and the available tropospheric humidity. We conclude that tropospheric temperature changes should be considered along with surface processes when assessing the causes of surface warming patterns.

## 1 Introduction

Rising greenhouse gas emissions will lead to climate warming on a global scale. However, the warming on regional to local scales is what directly affects people (Sutton et al., 2015). Therefore, adaptation and mitigation strategies must accord with regional climate change projections (Hall, 2014). Observations and climate simulations show that local warming deviates substantially from the global mean (Collins et al., 2013; Izumi et al., 2013; Good et al., 2015; King, 2019). Regionally amplified warming is especially relevant as it increases the impacts of climate change in affected regions. A prominent example of a regional warming amplification is the Arctic amplification, which is the strongest in the Northern Hemispheric cold season (Pithan and Mauritsen, 2014; Stuecker et al., 2018). In the mid-latitudes and tropics, land areas warm more than the ocean, as seen in observations and climate simulations (Byrne and O'Gorman, 2018; Chadwick et al., 2019). Within continents, several hot spots exhibit amplified warming compared to the surroundings (Diffenbaugh and Giorgi, 2012). Amplified surface warming will have far-reaching consequences, for example, increased heatwaves and droughts with resulting heat-related health impacts (Kovats et al., 2014; Son et al., 2019). Thus, there is a need to better understand the causes of such surface warming anomalies,

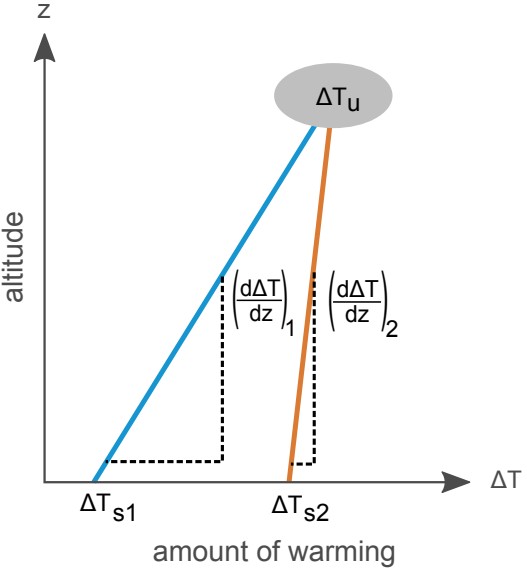

**Figure 1.** Geometrical illustration of how lapse-rate changes are related to surface warming. The blue line $(d\Delta T/dz)_1$ shows a large lapse-rate change, as expected over the ocean, and the orange line $(d\Delta T/dz)_2$ a small lapse-rate change $((d\Delta T/dz)_1 > (d\Delta T/dz)_2)$, as expected over land. We assume, that the warming in the upper troposphere $(\Delta T_u)$ is similar in both cases. From this assumption, it geometrically follows that the amount of surface warming is large $(\Delta T_{s2})$ where the lapse-rate change is small, and the surface warming is small $(\Delta T_{s1})$ where the lapse-rate change is large.

in order to assess whether climate models can reliably simulate these causes. Many hot-spots of amplified warming are arid or semi-arid areas. Such land surface warming hot-spots are often assumed to be primarily caused by changes in the partitioning of surface energy fluxes (Huang et al., 2017a, b; Barcikowska et al., 2020), but it is unclear if this surface perspective is sufficient in explaining the regional warming differences (Byrne and O'Gorman, 2013a; Berg et al., 2016; Koutroulis, 2019).

The amplified warming over land in comparison to oceans has been related to geographical variations in lapse-rate changes

in the troposphere (Joshi et al., 2008; Fasullo, 2010; Byrne and O'Gorman, 2013a, 2018). Lapse-rate changes in warmer climates are governed by the moist-adiabatic lapse rate, which decreases with warming. A decrease of lapse rates with warming is equivalent to a stronger tropospheric than surface warming or an increase in the atmospheric stability. Yet, the available humidity limits the magnitude of changes in lapse rates. As a result, lapse-rates over oceans, where moisture is abundant, are closer to the decreasing saturated moist-adiabatic lapse rate than over land. Whenever moisture is limited over land areas, the

influence of the temperature-independent dry-adiabatic lapse rate weakens lapse-rate changes. In the presence of a horizontally homogeneous upper-tropospheric warming, such differences in lapse-rate changes lead to stronger warming over land than ocean (Joshi et al., 2008; Fasullo, 2010; Byrne and O'Gorman, 2013a, 2018). We illustrate the differing tropospheric lapse-rate changes over land and ocean in Fig. 1, which provides a graphical explanation of the influence of lapse-rate changes on

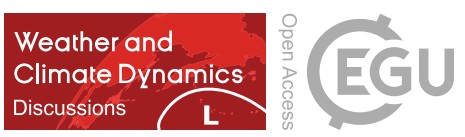

surface warming. Note that this figure does not imply causality. Assuming constant upper tropospheric warming, variations in
near-surface warming cause variations in lapse-rate changes, irrespective of the origins of these variations.

The influence of lapse-rate changes on the land-ocean warming contrast, displayed in Fig. 1, has been described mostly for tropical regions for two main reasons. First, the current tropical atmospheric stratification is governed by the moist-adiabatic lapse rate in all seasons (Xu and Emanuel, 1989; Schneider, 2007). Consequently, future changes in the moist-adiabatic lapse rates will be crucial. In summer, however, also the stratification in mid-latitudes seems to be governed by the moist-adiabatic
lapse rate (Korty and Schneider, 2007; Frierson and Davis, 2011; Zamora et al., 2016). The second reason is that tropical upper tropospheric horizontal temperature gradients are small as a result of the small Coriolis parameter (Charney, 1963, 1969; Sobel and Bretherton, 2000). Therefore, we expect greenhouse gas-driven upper tropospheric temperature changes to be horizontally homogeneous. In contrast, the spatial pattern of upper tropospheric warming to be expected outside the tropics is less clear.

In regards to variations in surface warming over land, lapse-rate changes have recently been suggested to exert a major
influence on the Mediterranean amplification (Kröner et al., 2017; Brogli et al., 2019a, b). The Mediterranean amplification describes enhanced warming in the summer season over land regions around the Mediterranean basin (Fig. 2a). Yet, we don't know if these findings are transferable to other land regions of the world that experience above-average surface warming.

In this article, we build upon the previous results to demonstrate how lapse-rate changes cause the Mediterranean amplification. We present novel idealized simulations that demonstrate this causality more clearly than our previous simulations.
Also, we show multi-model evidence for this causality. Additionally, we present evidence that lapse-rate changes are a driver for above-average land warming during the summer season in mid-latitudes and tropics across both the Northern and Southern Hemispheres. The findings originate from streamlined idealized simulations and the analysis of regional climate model (RCM) and global climate model (GCM) ensembles.

## 2 Materials and Methods

### 2.1 Analysis of CORDEX simulations over Europe

We analyze RCM simulations from the European Coordinated Regional Climate Downscaling Experiment (EURO-CORDEX) ensemble (Jacob et al., 2014, 2020). We use a total of 32 simulations performed with five different RCMs, driven by nine different GCM simulations from the Coupled Model Intercomparison Project Phase 5 (CMIP5) ensemble (Taylor et al., 2012). The EURO-CORDEX simulations feature horizontal resolutions of 0.44° or 0.11°. Further details can be found in Table A1.
All the downscaled GCM simulations assume the high-emission scenario RCP8.5 (Moss et al., 2010). We assess the 30-yr seasonal average temperature changes between 2070-2099 and 1971-2000.

### 2.2 Idealized simulations over Europe

To establish the causality between lapse-rate changes and the European summer warming, we perform simulations using the regional climate model COSMO-CLM version 4.8 (CCLM4.8), which is the model of the Consortium for Small-Scale




Modeling, COSMO (Baldauf et al., 2011) in climate mode (Rockel et al., 2008). The CCLM4.8 simulations feature a horizontal resolution of 0.44° (ca. 50 km), use 40 stretched vertical levels, and are covering the European domain following the EURO-CORDEX framework (Jacob et al., 2014, 2020). The simulations extend related experiments used in Kröner et al. (2017); Brogli et al. (2019a, b).

### 2.2.1 Characteristics of idealized simulations

The following list contains the key characteristics of the simulations performed:

– We perform regular transient regional climate simulations with CCLM4.8, where we downscale the two GCMs MPI-ESM-LR (Stevens et al., 2013) and HadGEM2-ES (Martin et al., 2011) and analyze the mean. To assess climate change, we select two time slices which we call CTRL and SCEN. CTRL is the 1971-2000 period and SCEN the 2070-2099 period assuming RCP8.5. FCC=SCEN−CTRL is used to quantify temperature changes, where the abbreviation FCC 80 stands for full climate change, and will be used in the remainder of the article.

– TD is the mean thermodynamic response of two idealized simulations where a vertically uniform warming profile is imposed at the lateral boundaries of CTRL. The shape of the vertical warming profile can be seen in Fig. 3f (gray line). The same profile is imposed on every lateral boundary gridpoint.

– TDLR is the second set of idealized simulations where both the large-scale thermodynamic and lapse-rate change are 85 imposed at the lateral boundaries of CTRL. The profile imposed on all boundary grid points is shown in Fig. 3d.

The idea behind the idealized experiments is the following: By comparing TD and TDLR we can quantify the influence of large-scale lapse-rate changes on the simulation result, as this is the only difference between the simulations. Simulated future changes that go beyond thermodynamic and lapse-rate changes, most prominently circulation and other dynamic changes, can be asessed by comparing TDLR and FCC. Figure A1 shows a comparison between circulation changes in TDLR and FCC and 90 confirms that circulation changes are absent in TDLR.

### 2.2.2 Technical implementation of idealized simulations

The imposed warming profiles in TD and TDLR are derived from the domain-mean warming of FCC and include a representation of the annual cycle of the warming (Brogli et al., 2019a, b). In TDLR, the domain-mean warming of FCC on every tropospheric model level is imposed, and constant warming is imposed above the approximate average height of the tropopause. 95 In TD, the mean warming of FCC, determined on the model level closest to 850 hPa, is imposed on all model levels. The annual cycle of SST changes is identically prescribed in TD and TDLR as a lower boundary condition and derived from the two-dimensional pattern of SST changes in FCC. For both atmospheric and ocean temperatures, the 30-yr daily mean changes from FCC are converted to the prescribed annual cycle using a spectral filter as described in Bosshard et al. (2011). When changing the temperature in TD or TDLR experiments, we change the humidity assuming constant relative humidity. Also, the 100 pressure is adjusted according to the hydrostatic balance (Schär et al., 1996). To quantify TD and TDLR, two different simu-





lations, based on either the climate change signal of MPI-ESM-LR or HadGEM2-ES have been performed and subsequently averaged for presentation. In the idealized experiments, the greenhouse-gas concentrations have been adapted to match SCEN (Kröner et al., 2017).

### 2.3 Analysis of CMIP6 simulations

For global analysis, we include state-of-the-art GCM simulations from the Coupled Model Intercomparison Project Phase 6 (CMIP6) (Eyring et al., 2016), but we also provide a comparison against the CMIP5 (Taylor et al., 2012) ensemble (in Fig. A2). We use CMIP6 simulations assuming the emission scenario SSP5-8.5 (O'Neill et al., 2016), which is close to RCP8.5. For the analysis, all CMIP6 simulations have been regridded to a common 1.4° grid. In all simulations, the 30-yr seasonal average temperature change between 2070-2099 and 1971-2000 is used to quantify climate change. A list of all CMIP6 simulations used is provided in Table A1.

Both the CMIP6 and EURO-CORDEX atmospheric data are available on pressure levels. Yet, to analyze lapse rate changes (K/m) the warming at the same geometrical height is more appropriate. For the quantification of lapse-rate changes we convert pressure levels to geometrical height.

## 3 Results & Discussion

### 3.1 European Summer Climate Change in the EURO-CORDEX ensemble

The Mediterranean amplification is a striking and robust feature in projections of the European summer climate (Fig. 2a) from the EURO-CORDEX RCM ensemble. Following the high-emission scenario RCP8.5, some Mediterranean areas warm up to 3 K more than the domain average (Fig. 2b) . In contrast to the Mediterranean, Northern European land areas typically exhibit below domain-mean summer warming (Fig. 2b). In the middle to upper troposphere, spatial variations in summer warming are negligible, as opposed to the surface (Fig. 2c). Regions characterized by small lapse-rate changes (Fig. 2d) in EURO-CORDEX coincide with regions of amplified surface warming (Fig. 2a,b). Figure 2 qualitatively indicates that different lapse-rate changes are connected to surface warming variations in Europe during summer.

### 3.2 Idealized Simulations for Europe

The causal effect that lapse-rate changes have on the Mediterranean amplification is demonstrated in Fig. 3, showing the three pairs of simulations FCC, TDLR, and TD (see Section 2.2). The near-surface warming anomaly of the regular downscaling simulations FCC (Fig. 3a), is in good agreement with EURO-CORDEX (Fig. 2b). From vertical profiles of the summer temperature change for FCC (Fig. 3b) we note that the warming is maximal at altitudes close to the tropopause. Also, in the upper troposphere, the warming is similar for Northern Europe and the Mediterranean. Yet, the lapse-rate changes are different over the Mediterranean and Northern Europe. In connection with a weak lapse-rate change, the surface warming in the Mediterranean is similar to the upper-tropospheric maximum and thus relatively high (Fig. 3b). Over Northern Europe, the lapse-rate

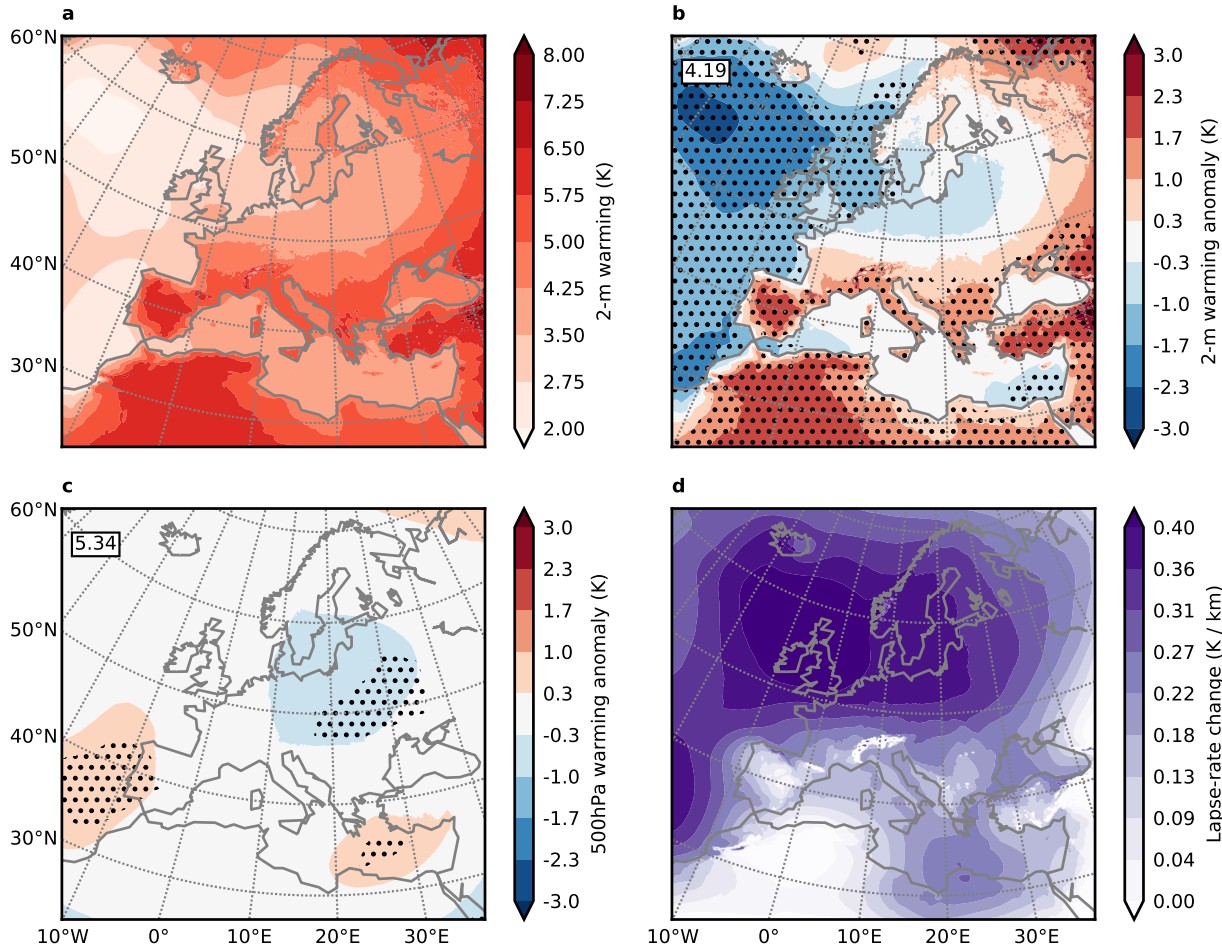

**Figure 2.** European summer warming and associated lapse-rate changes in the 0.11°-EURO-CORDEX ensemble for 1971-2000 versus 2070-2099, assuming the RCP8.5 emission scenario. The summer season is taken as June, July, and August (JJA). (a) Mean summer 2-m warming of 15 EURO-CORDEX simulations shown as absolute values. (b) The same 2-m warming shown in (a) but expressed as warming anomaly or deviation from the domain mean. Red colors denote above domain-mean warming and blue colors below domain-mean warming. The number in the upper left of the map shows the domain mean warming in K. Stippling shows regions where all simulations agree on the sign of the warming anomaly. Overall, (b) highlights the pattern of the warming shown in (a). (c) Same as in (b) but on 500 hPa. (d) Lapse-rate changes expressed as warming difference between 500 hPa and 850 hPa in K / km for the simulation ensemble shown in (a), (b), and (c).

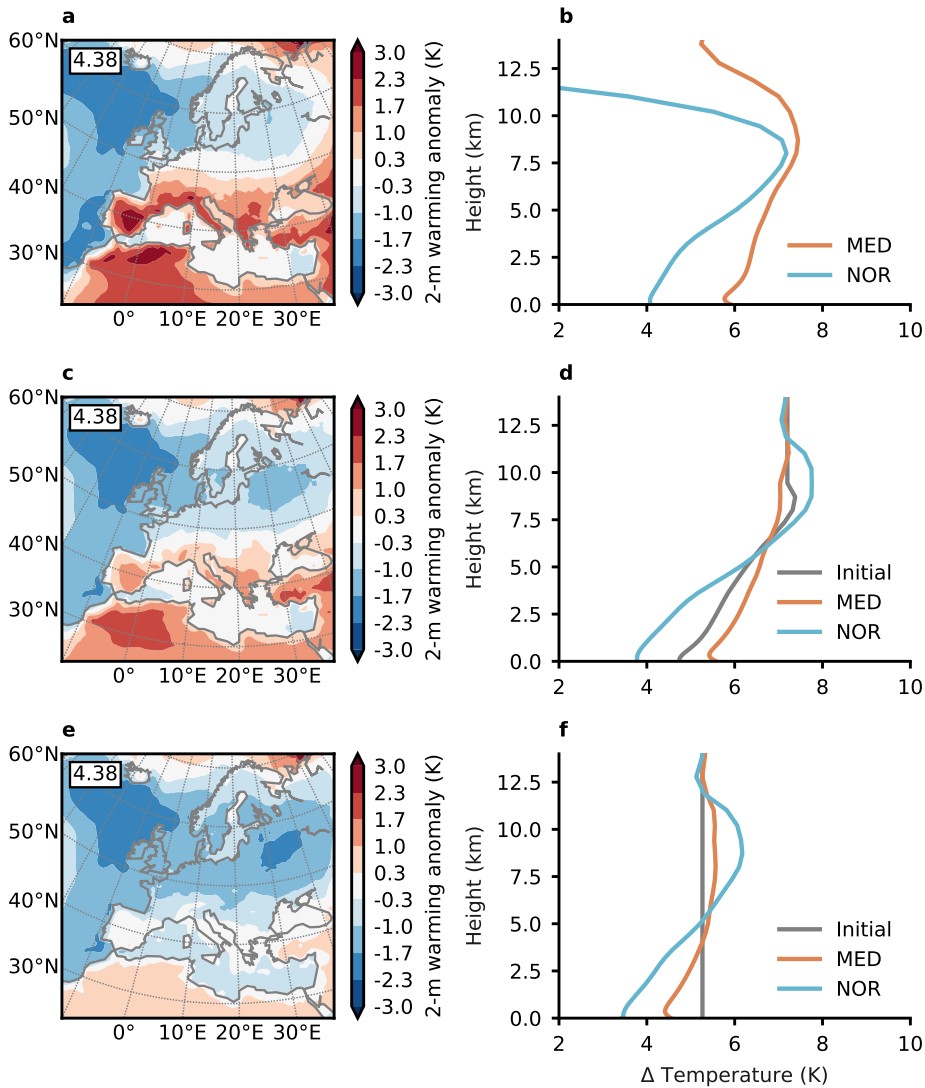

**Figure 3.** Idealized simulations demonstrating the importance of tropospheric lapse-rate changes for European surface summer (JJA) warming anomalies. (a) Mean summer 2-m warming anomaly (deviation from the mean given in the upper left) for FCC, which is the mean of two RCM simulations dynamically downscaling two different GCMs assuming RCP8.5. We assess the warming between 2070-2099 and 1971-2000. (b) Mean vertical profiles of the summer temperature changes for the simulations shown in (a) for Mediterranean (orange line) land grid points (MED; between 30° and 44° N) and Northern European (blue line) land grid points (NOR; between 55° N and 70° N). (c-d) Same as in (a-b), but for TDLR, which is the mean of two idealized simulations, where the mean vertical warming profile as shown by the gray line in (d) is imposed on all lateral boundary grid points of the RCM simulations for 1971-2000. (e-f) Same as (c-d) but for TD that differs from TDLR by the imposed warming profile shown by the gray line in (f). The domain-mean warming of FCC has been used to compute all anomalies shown in (a,c,e) for better comparability.





change is stronger and the surface warming smaller than over the Mediterranean. In general, the vertical warming profiles generated by the model (Fig. 3b) are very similar to the idealized picture shown in Fig. 1.

The Mediterranean amplification is reproduced by TDLR (Fig. 3c), the idealized simulations that are forced with a large-scale tropospheric lapse-rate change. In contrast, the Mediterranean amplification is absent in TD where we prescribe no

large-scale lapse-rate changes (Fig. 3e). To understand the reason for the difference between the two idealized simulations, we compare the respective vertical warming profiles (Fig. 3d,f). By design, the shape of the initial warming profile imposed at the lateral boundaries (shown by the gray lines in Fig. 3d,f) differs between the two simulations. In TDLR (Fig. 3d), the lapse-rate change in the Mediterranean dynamically weakened compared to the imposed profile and the surface warming increased as a consequence (Fig. 3d, orange vs. gray line). In Northern Europe, we see the opposite, meaning that the lapse-rate change

simulated in the domain interior is stronger than what was prescribed, and the surface warming has decreased in response (Fig. 3d, blue vs. gray line). In other words, for the Mediterranean, some of the warming imposed at the lateral boundaries has been dynamically re-distributed from the upper troposphere to the lower troposphere, and vice-versa for Northern Europe.

In the TD experiments, where we impose vertically uniform warming at the lateral boundaries, the surface warming in both the Mediterranean and Northern Europe is lower than what was imposed at the lateral boundaries (Fig. 3f) (Lenderink et al.,

2019). This follows from similar dynamic alterations of the imposed warming profiles as in TDLR. Also in TD, simulated lapse-rate changes are larger in Northern Europe than the Mediterranean (Fig. 3f). Yet, the regional maximum in Mediterranean surface warming doesn't appear in the absence of an upper tropospheric warming maximum, even though we impose a comparatively high surface warming (Fig. 3f) and the land-surface feedbacks in the model are fully interactive.

Summarizing Fig. 3, we find that two decisive changes in the climate system are needed to simulate the Mediterranean

amplification. First, a strong and horizontally uniform large-scale upper tropospheric warming must be present. Second, the lapse-rate changes are dynamically altered depending on the location and directly connected to the extent of summer surface warming. Since the Mediterranean is dryer in summer than Northern Europe (Brogli et al., 2019a), the dynamic alteration of the lapse-rate change profile supports the idea that moisture availability controls the strength of the lapse-rate changes in mid-latitudes during summer (Joshi et al., 2008; Fasullo, 2010; Byrne and O'Gorman, 2013a, b, 2018; Brogli et al., 2019a). The

simulated local adjustment of lapse-rates across the troposphere suggests that the European summer atmosphere is vertically mixed by local convection and subsidence. Lapse-rate changes are thus connected to the decrease of the moist-adiabatic lapse rate at warmer temperatures and radiative-convective equilibrium (Held and Soden, 2000). Small lapse-rate changes may occur where moist-adiabatic vertical mixing is inhibited by dry conditions and temperature-independent dry-adiabatic mixing plays a larger role.

It may be surprising that the middle to upper tropospheric temperature change in summer over Europe is spatially uniform. While this seems clear in the simulations we analyzed, the physical reasons behind it remain more speculative. Generally, a strong upper tropospheric summer warming can be expected, since most of the globe is covered by oceans where one would expect strong lapse-rate changes due to the abundance of moisture. The uniformity of the warming could be related to the weak equator-to-pole temperature gradient in summer, which also results in weak baroclinicity, implying that warming gradients will

also be small. Also, the long 30-yr periods that we averaged to obtain the results might act to smooth warming gradients.

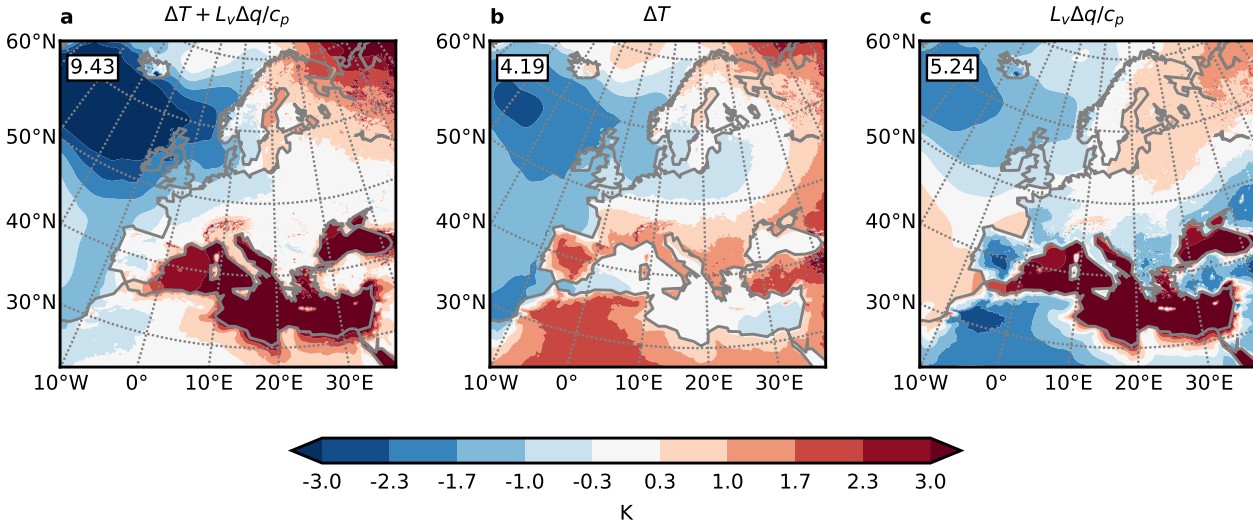

**Figure 4.** Decomposition of the anomaly of surface moist enthalpy ($H$) changes in the 0.11° EURO-CORDEX ensemble for the summer season. (a) Overall anomaly (deviation from the domain mean) of the change in moist enthalpy, normalized by the specific heat of air ($c_p$) to yield the unit Kelvin. Thus, the surface moist enthalpy change has been computed as $\Delta H = \Delta T + L_v \Delta q / c_p$. The anomalies of the two components of $H$ are shown separately. (b) shows the anomaly of $\Delta T$, while (c) shows the anomaly of $L_v \Delta q / c_p$.

## 3.3 Changes in moist enthalpy in EURO-CORDEX

Previously, we argued that the differential importance of moist- and dry-adiabatic vertical mixing controls the magnitude of summer lapse-rate changes and regulates the surface warming. A similar argument can be made when analyzing the change in moist enthalpy (Berg et al., 2016; Byrne and O'Gorman, 2018; Matthews, 2018). The moist enthalpy is given by $H = c_p T + L_v q$, where $c_p$ is the specific heat of air, $T$ is temperature, $L_v$ the latent heat of vaporization, and $q$ the specific humidity. In a warming climate, the change in moist enthalpy quantifies the combined effect of changes in internal energy given by temperature changes ($c_p \Delta T$) and the change in latent energy, which could be released by condensation if air was lifted to the top of the atmosphere ($L_v \Delta q$).

Figure 4 shows the spatial anomalies of the overall change in moist enthalpy ($\Delta H$) and the two components $c_p \Delta T$ and $L_v \Delta q$ seperately. The data is from the EURO-CORDEX ensemble and the summer season (JJA). Note that over land, the simulations project a spatially relatively homogeneous change in $H$ (Fig. 4a). This means that total amount energy used for raising temperatures or humidity is similar troughout the European continent. Yet, the change in internal energy or temperature shows a regional maximum the Mediterranean (Fig. 4b) while the change in latent energy is smaller in the Mediterranean than elsewhere in the domain (Fig. 4c). The change in latent energy describes the potential for latent energy release by convection, which is also the root of lapse-rate changes. Thus, the change in moist enthalpy supports the notion that the additional available





energy connected to climate change in the Mediterranean rather translates to an increase in surface temperature ($c_p\Delta T$) than to an increase in convective latent heat release ($L_v\Delta q$) and vice-versa for Northern Europe. The below-average increase in $\Delta q$ (Fig. 4c) is a clear sign for limited moisture availability in the Mediterranean, because from the climatological temperatures alone, one would expect a above-average increase of $\Delta q$ in the relatively warm Mediterranean (the warmer air could potentially carry more water vapour).

### 3.4 Statistical analysis of EURO-CORDEX simulations

The results shown in Figs. 2, 3 and 4 suggest that lapse-rate changes are crucial for determining European summer surface warming anomalies in the multi-model mean in EURO-CORDEX and our RCM simulations. By statistically analyzing lapse-rate changes in 32 single members of the EURO-CORDEX ensemble, we here confirm the robustness of these findings and the transferability to different RCMs. We consider the full ensemble (Table A1) and a subset of 14 simulations performed with the RCM RCA4 (Strandberg et al., 2014) that use identical parameterization schemes and only differ in the large-scale forcing from different GCMs (Table A1).

We show linear regressions in Fig. 5. Previously, we identified two core triggers for the Mediterranean amplification. First, the local lapse-rate changes over Northern Europe must be larger than over the Mediterranean. Second, a strong homogeneous upper tropospheric warming must be present. The linear regressions support this idea (Fig. 5), showing that, first, the larger the difference in lapse-rate changes between Northern Europe and the Mediterranean, the larger the Mediterranean amplification (Fig. 5a-b). Second, the larger the domain-mean lapse-rate change (i.e. more upper tropospheric warming compared to the surface), the larger the Mediterranean amplification (Fig. 5c-d). All the positive correlations found in Fig. 5 are statistically significant ($p < 0.0006$) and feature $R^2$ values ranging from 0.46 to 0.93. Thus, Fig. 5 supports the notion that in a variety of climate projections, mean lapse-rate changes and spatial differences in lapse-rate changes regulate the magnitude of summer surface warming.

However, a remaining open question from Fig. 5 is what intrinsic differences between the simulations cause the different lapse-rate changes in the EURO-CORDEX ensemble members, especially in the ensemble using the same RCM. A limited body of research suggests that such intermodel differences might be related to regional SST warming differences in GCMs (Po-Chedley et al., 2018; Tuel, 2019). Different lapse-rate changes between models have also been suggested to be connected to climate sensitivity (Ceppi and Gregory, 2017), which makes this question even more relevant.

The fact that lapse-rate changes are relevant in the context of European summer climate change raises the question of whether they are equally influential in other regions of the planet. To this end, we further explore surface warming anomalies and the connected lapse-rate changes in global climate simulations.

### 3.5 Analysis on Global Scale

Figure 6 shows end-of-century summer lapse-rate changes for all land regions of the world within the mid-latitudes and tropics (-66° S < latitude < 66° N) from the CMIP6 global simulation ensemble. Additionally, we verified the ability of CMIP6 models to reproduce the lapse-rate changes over Northern Europe and the Mediterranean simulated by RCMs (Fig. 7). Generally,

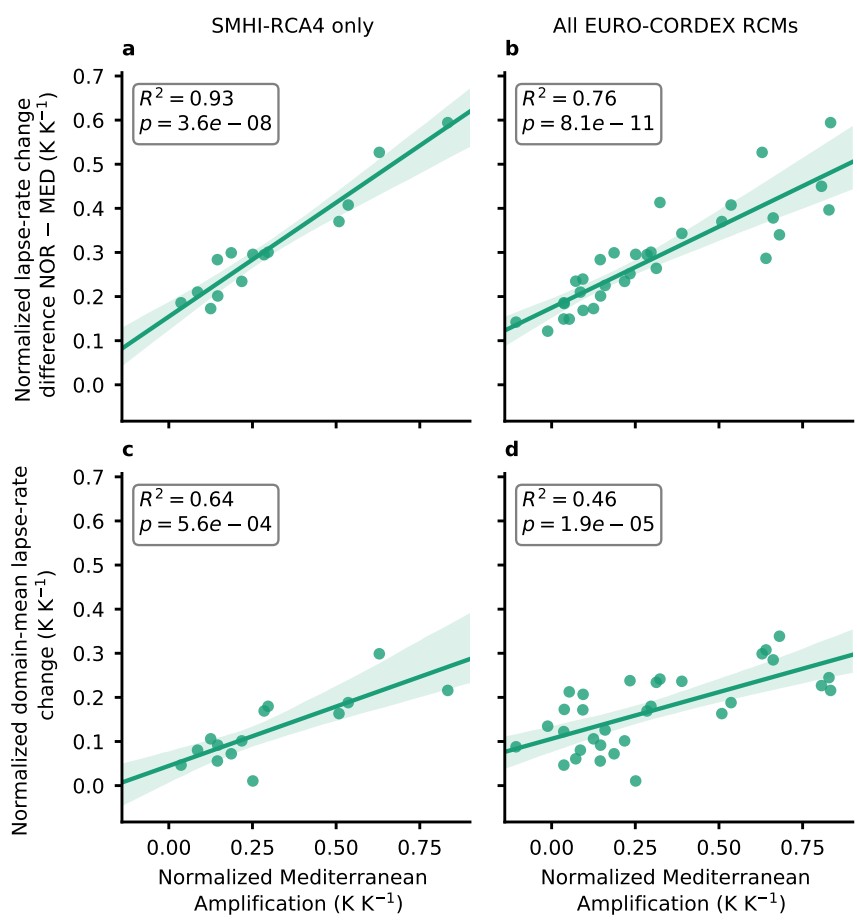

**Figure 5.** Linear regression linking the enhanced Mediterranean summer 2-m warming to lapse-rate changes in the EURO-CORDEX ensemble. (a,b) Regression linking the Mediterranean amplification and differences in lapse-rate changes between Northern Europe (NOR) and the Mediterranean (MED) over land. (c,d) Regression linking the Mediterranean amplification to the European-scale domain-mean lapse-rate change. (a,c) A subset of the EURO-CORDEX ensemble using one single regional climate model, namely SMHI-RCA4. (b,d) All EURO-CORDEX simulations that were used in this study (Table A1). Each dot represents a simulation with a horizontal resolution of 0.11° or 0.44°. The figure is based on summer-mean (JJA) changes between 1971-2000 and 2070-2099, assuming RCP8.5. We normalized both the Mediterranean amplification and lapse-rate changes with the domain-mean warming, yielding dimensionless values (K/K). The lapse-rate changes have been computed as the difference between the 500 hPa warming and the 850 hPa warming. We evaluate the Mediterranean between 30° and 44° N and Northern Europe between 55° N and 70° N. The boxes in the upper left corner of the panels show the statistics of the linear regression fit, while the shadings show the 95% confidence interval of the linear regression estimated by 1000-fold bootstrapping.

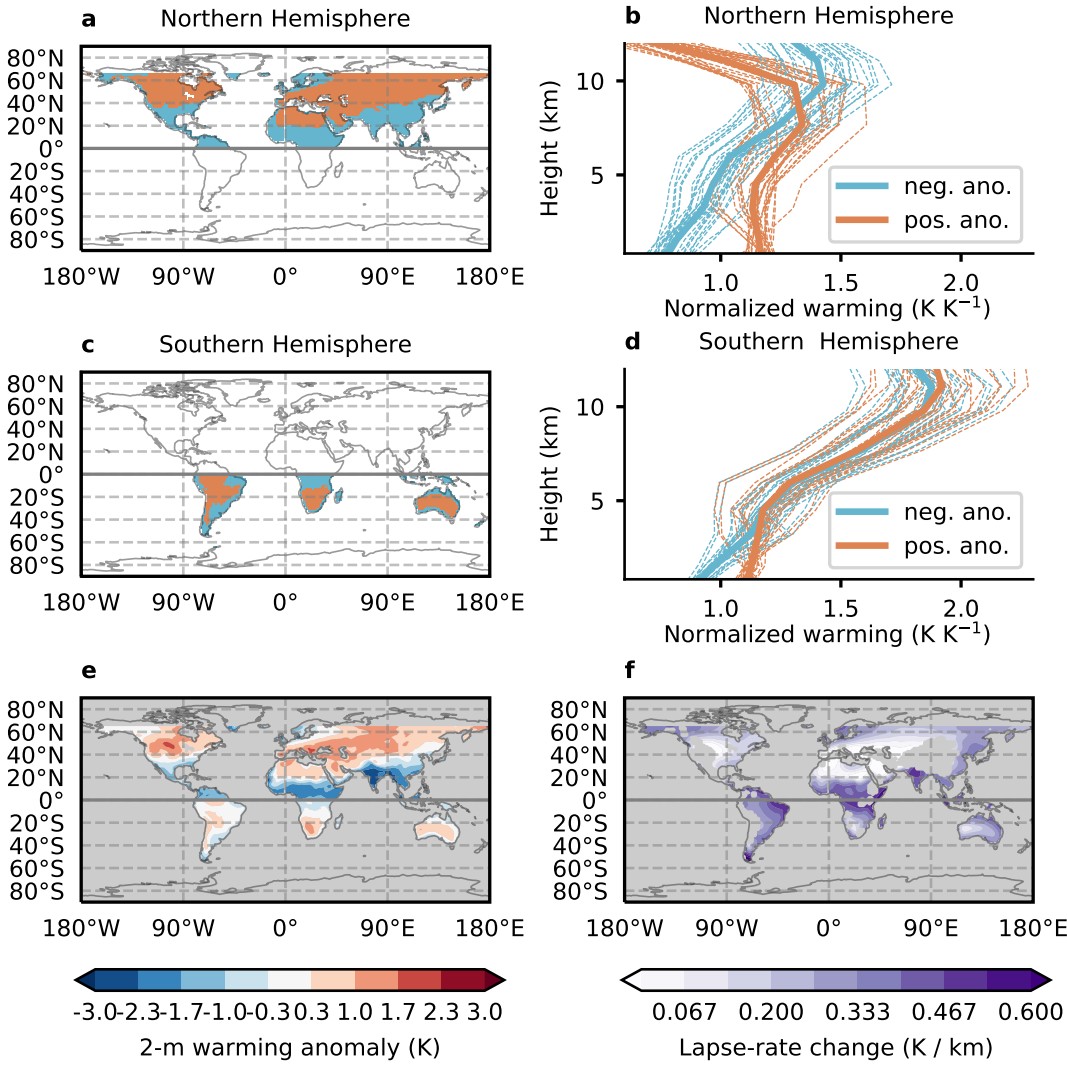

**Figure 6.** Summer lapse-rate changes as projected by CMIP6 simulations. The changes are evaluated between 1971-2000 and 2070-2099, assuming the SSP5-8.5 scenario for JJA north of the Equator and December, January, and February (DJF) south of the Equator. (a) Map showing areas where the ensemble mean Northern Hemispheric summer near-surface warming is above (orange) and below (blue) average over land in the mid-latitudes and tropics (0° - 66° N). (b) Mean vertical profiles of summer warming in regions where the surface-warming is above and below average (orange and blue profiles, respectively). The thin dashed lines show individual ensemble members, and the bold line shows the ensemble mean. The vertical warming profiles are normalized by every simulation's mean summer warming over land grid points on 925 hPa. (c) Same as (a) for the Southern Hemisphere (0° - 66° S). (d) Same as (b) for the areas shown in (c). (e) Map of mean land 2-m warming anomalies in the Northern and Southern Hemispheres during the respective summer season. Areas masked in the analysis are shown in gray. Red colors show above-average warming and blue below-average warming. (f) Map of lapse-rate changes evaluated as warming difference between 500 hPa and 850 hPa (K/km). Masked areas (sometimes due to high topography) are shown in gray.

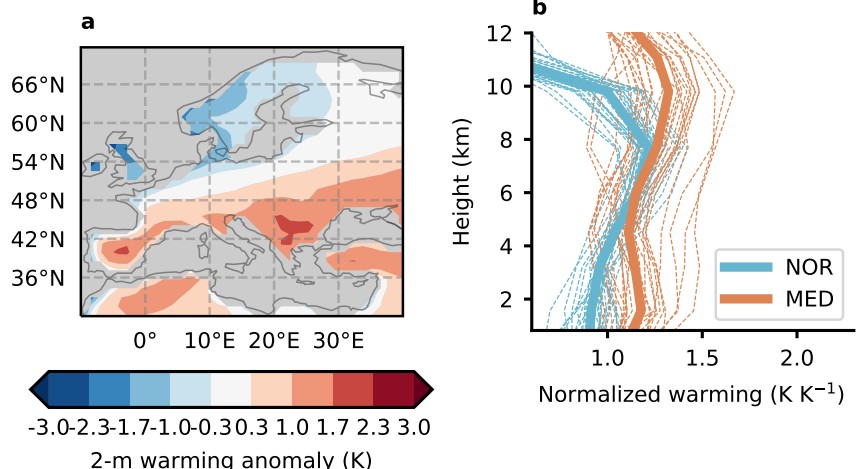

**Figure 7.** The Mediterranean amplification in CMIP6 simulations. (a) Ensemble mean summer near-surface warming anomaly with respect to the ensemble mean of Northern Hemispheric land grid points (0°-70° N). (b) Mean vertical warming profiles over the Mediterranean (MED; orange) and Northern Europe (NOR; blue) in summer. The thin dashed lines show individual ensemble members and the bold line shows the ensemble mean. The MED is evaluated between 30° N, 44° N, 10° W and 40° E, while NOR is evaluated between 55° N, 70° N, 0° E and 40° E and only land grid points are used. The warming has been normalized by the mean warming on 925 hPa over NOR and MED.

the CMIP6 models considered also exhibit the Mediterranean amplification (Figs. 7 and 6e) and show pronounced regional
differences in lapse-rate changes.

The global analysis confirms that weak lapse-rate changes are connected to enhanced land-surface warming across the Northern and Southern Hemispheres in the summer season. During the Northern Hemispheric summer, the warming on an altitude of ∼8 km is similar in all regions (Fig. 6b). Regions characterized by above-average summer warming (Fig. 6a) robustly show comparably small lapse-rate changes (Fig. 6b). Lapse-rate changes in areas with below-average warming (Fig.
6a) are large (Fig. 6b). Above-average summer warming in the Northern Hemisphere is projected in Central North America, the Mediterranean, Northern Africa, the Middle East, and large parts of Central Asia (Fig. 6a,e). In contrast, CMIP6 models show below-average summer land warming and large lapse-rate changes in the Tropics, Northern Europe, and in high-latitude North America (Fig. 6a,e,f), consistent with the analyses over Europe.

In the Southern Hemisphere, the summer lapse rate changes, and the resulting surface warming anomalies are in line with
previous findings (Fig. 6c-d). Lapse-rate changes are stronger in regions exhibiting below-average warming in comparison to regions with above-average warming, which is robust in all simulations considered (Fig. 6d). Thus, lapse-rate changes are closely related to the pattern of summer temperature change, also in the Southern Hemisphere. One remarkable difference is that the average altitude at which the warming starts to be spatially homogeneous occurs at ∼4 km in the Southern Hemisphere, which is lower than in the Northern Hemisphere, where it is located at ∼8 km (Fig. 6b,d). Also, surface warming anomalies are





less pronounced (Fig. 6e). We assume that this is because the fraction of oceans is much larger in the Southern Hemisphere, which reduces the altitude up to which land surface inhomogeneities can influence temperature changes. In the Southern Hemisphere, the regions connected to above-average land warming are continental areas in South America, South Africa, and Australia. Below-average warming is found along the coast of the previously mentioned regions and New Zealand (Fig. 6c,e). Disparities in lapse-rate changes between continental and coastal areas (Fig. 6f) are a further indication that the moisture
availability influences the strength of lapse-rate changes as regions along the coast tend to be more humid than continental regions.

The findings presented for the CMIP6 ensemble in Fig. 6 are in agreement with the results obtained using the CMIP5 ensemble (Fig. A2), although the magnitude of surface warming anomalies has increased from CMIP5 (RCP8.5) to CMIP6 (SSP5-8.5). In summary, the qualitative analysis shown in Fig. 6 suggests that the findings for the European region (Figs. 2-5)
are likely transferable to other regions on the globe, and future more specific regional research is desirable.

## 4 Conclusions

We find that variations in the vertical warming with climate change, given by atmospheric lapse-rate changes, are decisive for the understanding of enhanced or reduced regional greenhouse-gas-driven surface warming during summer. We provide additional evidence for this finding in Europe, where we demonstrate that both a strong large-scale upper tropospheric warming
and regionally modified lapse-rate changes are key reasons for the Mediterranean amplification. The results are consistent across CORDEX and CMIP climate model ensembles. Additionally, we showed that lapse-rate changes are closely connected to summer surface warming anomalies on a global scale over land areas. The connection between lapse-rate changes, which are governed by the well-understood temperature dependence of the moist-adiabatic lapse rate, and summer warming anomalies, increases our confidence that climate models can accurately project such anomalies.

Our results suggest that lapse-rate changes have more extensive consequences than previously thought. First, lapse-rate changes have a decisive influence on the summer-season warming pattern in mid-latitudes and the tropics. Second, lapse-rate changes influence surface warming patterns within land regions in addition to the land-ocean warming contrast. Tropospheric temperature and lapse-rate changes should thus be considered when assessing the causes for surface warming anomalies, in addition to surface processes. Since evidence from multiple studies points to a connection between surface moisture and lapse-
rate changes (Joshi et al., 2008; Byrne and O'Gorman, 2013a, b, 2018; Brogli et al., 2019a, this study), it might be possible to use surface moisture as an observational constraint for future summer warming. Our study is solely based on climate-change simulations. Therefore, an open task is to diagnose historical differences in lapse-rate changes from observations. It is, however, hard to estimate what level of warming is necessary for lapse-rate changes to be detected in observations, as this requires multiple long records of homogeneous upper-tropospheric measurements, which are sparse and prone to biases (Santer
et al., 2005; Stickler et al., 2010; Thorne et al., 2011; Flannaghan et al., 2014; Santer et al., 2017).



*Code and data availability.* The codes that are the basis for our idealized experiments are open source and can be obtained from https://www.doi.org/10.5281/zenodo.4890235 or https://github.com/broglir/pgw-python. The weather and climate model CCLM is free of charge for research applications (for more details see: http://www.cosmo-model.org). EURO-CORDEX, CMIP6 and CMIP5 data are available from ESGF nodes (https://esgf-data.dkrz.de/projects/esgf-dkrz).

*Author contributions.* R.B., S.L.S., N.K. and C.S designed research, R.B. performed simulations, R.B. and N.K. analyzed data, R.B., S.L.S. and C.S wrote the article.

*Competing interests.* The authors declare no competing interests.

*Acknowledgements.* We acknowledge PRACE for awarding us access to Piz Daint at Swiss National Supercomputing Center (CSCS, Switzerland). Furthermore, we acknowledge the COSMO, CLM and C2SM communities for developing and maintaining the RCM. We
also acknowledge the World Climate Research Programme's Working Groups on Regional Climate and on Coupled Modeling, responsible for CORDEX and CMIP, and we thank the climate modeling groups for producing and making available their model output. We thank Urs Beyerle for preparing the CORDEX and CMIP data for our analysis and Paul O'Gorman for discussions on the topic of the article. This project has received funding from the European Union's Horizon 2020 research and innovation programme under grant agreement No 820829 and from the Swiss National Science Foundation under number 192133.





**Table A1.** Simulations used for analysis. The original simulations are regional climate modeling (RCM) experiments as described in Section 2.2, and the additional RCM simulations are from the EURO-CORDEX ensemble. Furthermore, global climate (GCM) simulations from the CMIP6 ensemble are used. Columns show the model name, the global driving simulation for RCMs, the realization for GCMs and the horizontal resolution of the atmospheric model. Note that the table spans multiple pages.

| Model name | Driving simulation (RCM) / Realisation (GCM) | Horizontal Resolution |
|---|---|---|
| **Original simulations** | | |
| CCLM4.8 | HadGEM2-ES | 0.44° |
| CCLM4.8 | MPI-ESM-LR | 0.44° |
| **EURO-CORDEX** | | |
| SMHI-RCA4 | EC-EARTH | 0.11° |
| SMHI-RCA4 | IPSL-CM5A-MR | 0.11° |
| SMHI-RCA4 | HadGEM2-ES | 0.11° |
| SMHI-RCA4 | MPI-ESM-LR | 0.11° |
| SMHI-RCA4 | NorESM1-M | 0.11° |
| SMHI-RCA4 | EC-EARTH | 0.44° |
| SMHI-RCA4 | IPSL-CM5A-MR | 0.44° |
| SMHI-RCA4 | HadGEM2-ES | 0.44° |
| SMHI-RCA4 | MPI-ESM-LR | 0.44° |
| SMHI-RCA4 | NorESM1-M | 0.44° |
| SMHI-RCA4 | CanESM2 | 0.44° |
| SMHI-RCA4 | CSIRO-Mk3-6-0 | 0.44° |
| SMHI-RCA4 | MIROC5 | 0.44° |
| SMHI-RCA4 | GFDL-ESM2M | 0.44° |
| KNMI-RACMO22E | EC-EARTH | 0.11° |
| KNMI-RACMO22E | IPSL-CM5A-MR | 0.11° |
| KNMI-RACMO22E | HadGEM2-ES | 0.11° |
| KNMI-RACMO22E | MPI-ESM-LR | 0.11° |
| KNMI-RACMO22E | NorESM1-M | 0.11° |
| KNMI-RACMO22E | EC-EARTH | 0.44° |
| KNMI-RACMO22E | HadGEM2-ES | 0.44° |
| CLMcom-CCLM4.8 | EC-EARTH | 0.11° |
| CLMcom-CCLM4.8 | HadGEM2-ES | 0.11° |
| CLMcom-CCLM4.8 | MPI-ESM-LR | 0.11° |
| CLMcom-CCLM4.8 | MPI-ESM-LR | 0.44° |
| CLMcom-CCLM5.0 | EC-EARTH | 0.44° |





| | | |
|---|---|---|
| CLMcom-CCLM5.0 | MIROC5 | 0.44° |
| CLMcom-CCLM5.0 | HadGEM2-ES | 0.44° |
| CLMcom-CCLM5.0 | MPI-ESM-LR | 0.44° |
| GERICS-REMO2015 | MPI-ESM-LR | 0.11° |
| GERICS-REMO2015 | NorESM1-M | 0.11° |
| MPI-CSC-REMO2009 | MPI-ESM-LR | 0.44° |
| **CMIP6** | | |
| ACCESS-CM2 | r1i1p1f1 | 1.875° x 1.25° |
| ACCESS-ESM1-5 | r1i1p1f1 | 1.875° x 1.25° |
| AWI-CM-1-1-MR | r1i1p1f1 | 0.9375° |
| BCC-CSM2-MR | r1i1p1f1 | 1.125° |
| CAMS-CSM1-0 | r1i1p1f1 | 1.125° |
| CanESM5 | r1i1p1f1 | 2.8° |
| CESM2 | r1i1p1f1 | 0.9° x 1.25° |
| CESM2-WACCM | r1i1p1f1 | 0.9° x 1.25° |
| CIESM | r1i1p1f1 | 1° |
| CNRM-CM6-1 | r1i1p1f2 | 1.4° |
| CNRM-ESM2-1 | r1i1p1f2 | 1.4° |
| EC-Earth3 | r1i1p1f1 | 0.7° |
| EC-Earth3-Veg | r1i1p1f1 | 0.7° |
| FGOALS-f3-L | r1i1p1f1 | 1° |
| FGOALS-g3 | r1i1p1f1 | 2° |
| FIO-ESM-2-0 | r1i1p1f1 | 1.875° x 0.625° |
| GFDL-CM4 | r1i1p1f1 | 1° |
| GFDL-ESM4 | r1i1p1f1 | 1° |
| GISS-E2-1-G | r1i1p1f2 | 2.5° x 2° |
| HadGEM3-GC31-LL | r1i1p1f3 | 1.875° x 1.25° |
| INM-CM4-8 | r1i1p1f1 | 2° x 1.5° |
| INM-CM5-0 | r1i1p1f1 | 2° x 1.5° |
| IPSL-CM6A-LR | r1i1p1f1 | 2.5° x 1.25° |
| MIROC6 | r1i1p1f1 | 1.4° |
| MIROC-ES2L | r1i1p1f1 | 2.8° |
| MPI-ESM1-2-HR | r1i1p1f1 | 0.9375° |
| MPI-ESM1-2-LR | r1i1p1f1 | 1.875° |
| MRI-ESM2-0 | r1i1p1f1 | 1.125° |
| NESM3 | r1i1p1f1 | 1.875° |






| NorESM2-LM | r1i1p1f1 | 2° |
| NorESM2-MM | r1i1p1f1 | 1.25° x 0.9375° |
| UKESM1-0-LL | r1i1p1f2 | 1.875° x 1.25° |



**Figure A1.** Changes in the mean summer zonal winds in idealized simulations forced by thermodynamics and lapse-rate changes (named TDLR) compared to fully transient climate simulations (named FCC). The first and third row show CCLM4.8 simulations over Europe with climate changes derived from HadGEM2-ES. The second and fourth row show CCLM4.8 simulations based on MPI-ESM-LR. The upper two rows show simulations that have been forced with domain mean lapse-rate changes and SSTs only (TDLR). The lower two rows show the corresponding transient climate simulations (FCC). (left column) Meridional crossection of isothermes (gray contours every 10 K) and zonal wind (purple contours in m/s) in the historical simulation (1971-2000). (middle column) same as the left column but for the future climate state (2070-2099). (right column) Difference in zonal wind between the historical and future simulation.



**Figure A2.** Same as Figure 6 but for an ensemble of CMIP5 simulations consisting of the following models: ACCESS1-0, ACCESS1-3, BCC-CSM1-1, BCC-CSM1-1-m, BNU-ESM, CanESM2, CESM1-BGC, CESM1-CAM5, CESM1-CAM5-1-FV2, CESM1-WACCM, CMCC-CESM, CMCC-CM, CMCC-CMS, CNRM-CM5, CSIRO-Mk3-6-0, FGOALS-g2, FIO-ESM, GFDL-CM3, GFDL-ESM2G, GFDL-ESM2M, GISS-E2-H, GISS-E2-H-CC, GISS-E2-R, GISS-E2-R-CC, HadGEM2-AO, HadGEM2-CC, HadGEM2-ES, INM-CM4, IPSL-CM5A-LR, IPSL-CM5A-MR, IPSL-CM5B-LR, MIROC5, MIROC-ESM, MIROC-ESM-CHEM, MPI-ESM-LR, MPI-ESM-MR, MRI-CGCM3 and MRI-ESM1.



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
