# Peer review of "Future summer warming pattern under climate change is regulated by lapse-rate changes"

_Weather and Climate Dynamics, 2021_

## Author Comment (AC3)

**Response to Referee #2 for wcd-2021-34**

The referee's comments are shown in black and italic. Our responses are shown in dark blue. When we paste text from the manuscript to the response, it is shown in dark red. For convenience we append to the end of the document a track-change version of our manuscript where one can quickly spot all changes we made to the manuscript.

***Review for "Future summer warming pattern under climate change is regulated by lapse-rate changes"***

*In this paper, the authors argue that lapse rate changes are fundamental to understanding the pattern of summertime warming projected in a variety of regional and global climate models. The authors present results from idealized modelling experiments to support their hypothesis that lapse rate changes are fundamental to understanding the pattern of surface warming across the globe and that lapse rate changes are a major factor that generate the pattern of surface warming that manifests across climate models.*

*The paper is well written and clear, and I think the investigation into the role of the atmosphere's role in generating land surface warming patterns is extremely important. However, in its current state, I don't think this paper provides an adequate explanation of the physics behind the creation of the amplification pattern over the Mediterranean; this makes the global analysis presented later in the paper less insightful. I was left wanting more evidence from the initial simulations to bolster the author's argument that the atmosphere's adjustment is crucial to understanding the warming pattern of summertime surface temperature.*

We thank the referee for taking the time for a detailed review and the constructive feedback on our manuscript.

***Major Comments***

*The authors allude (lines 39-40) to the fact that the spatial warming pattern is at least partially driven by energy partitioning between latent and sensible at the land surface. To me, this is a null hypothesis, and the authors must use their experiments to quantify the role of the atmosphere in amplifying the pattern of warming initially generated by the local differences in energy flux partitioning at the land surface. In my mind the authors need to show that convective activity or circulation adjustment to the warming pattern driven by energy flux partitioning near the land surface is amplified by the lapse-rate adjustment accomplished by the atmosphere for their conclusions to be valid.*

*To me, the analysis of novel experiment done by the authors (application of a vertically uniform boundary condition) does not prove that lapse rate changes are the ultimate cause of the surface warming pattern because there is no evidence presented about the dynamical adjustment. As a reader, I would like to know how the retribution of energy*

*that amplifies the warming in the Mediterranean is accomplished in the TDLR simulations (and not accomplished in the TD simulations). Could it be that an unrealistic (vertically uniform) boundary condition destabilizes the atmosphere, causing enhanced convection across all of Europe and a large dynamical adjustment that mixes out some component of the summertime warming pattern? Or is it that the realistic lapse rate boundary condition applied to the TDLR simulations induces large scale subsidence over the Mediterranean and amplifies the local warming? In any case, I was left wanting an analysis of the atmospheric motions that give rise to the lapse rate changes shown in Figs. 3d and f. The authors approach this analysis in Fig. 4, but the moist enthalpy changes could also be driven by the surface flux partitioning; without a complementary analysis of the atmospheric adjustment, I think the argument is much less convincing.*

*In particular, the presentation of Fig. 3e needs to be changed. To my eye, removing the TD simulation domain mean (rather than the FCC domain mean) would reveal a pattern similar (but not exactly the same) as the TDLR experiment, suggesting that energy flux partitioning at the land surface is fundamentally responsible for the warming gradient and only accentuated by the atmospheric response.*

We think that you raise a very relevant and interesting point. First, regarding Figure 3 we fully agree and did make the suggested adjustment to the figure, which is detailed also in an answer to a minor comment below. There is indeed an amplification of the warming also in TD, which we discuss in detail in the text (lines 149-157 in the revised manuscipt). We agree that the near-surface conditions (especially the moisture availability) do exert an important influence on the summer warming pattern. Yet from the profiles shown in Figure 3 it is also clear, that the surface influences the entire troposphere and vice-versa. This is now stated in the text (staring on line 215):

Overall, during European summer, the entire troposphere is relevant to understand the surface warming pattern: From TD we can observe that surface moisture gradients influence the warming up to 10-12 km height (Figure 3i). On the other hand, the extra upper tropospheric warming introduced in TDLR strongly affects near-surface warming levels (Figure 3f).

As suggested by the referee, we added an analysis of vertical motions within our simulation domain. This is shown in the newly added Figure 5. In summary we find that in our idealized simulations dynamical changes are insignificant, especially when compared to the fully GCM-driven run. This also agrees with Figure A1. This allowed us to state with more certainty that the adjustments of tropospheric lapse-rates are achieved by the warming/moistening of the atmosphere rather than from altered atmospheric dynamics. We now provide a much more extensive physical interpretation of the simulation results (lines 167-217), highlighting the importance of moisture availability for lapse-rate changes to occur. Also we argue using the thermodynamic equation that stratification changes can lead to warming in the absence of changes in vertical motions (lines 193-203). The moisture availability is also shown in the new Figure 4. Pasting the entire additional description of our

experiments in this response would be to long but we add the new summarizing section from our manuscript below, which hopefully well addresses your concerns:

Bringing together the results from Figs. 3, 4, and 5 we interpret the results as follows: Warming which we artificially impose in the idealized simulations is vertically re-distributed throughout the troposphere. The vertical motions required to achieve the re-distribution of the warming are already present during summer in the CTRL simulation (Fig. 5). The warming and moistening of the atmosphere in the TD simulation suffices to induce lapse-rate changes. These lapse-rate changes are more pronounced in present-day moist regions than in dry regions (Fig. 3i and Fig. 4a) because the vertical mixing follows the temperature-dependent moist-adiabatic lapse rate more frequently in moist regions. While in both TD and TDLR the lapse-rates locally adjust according to the near-surface humidity and in both simulations lead to an amplification of the Mediterranean warming, the very high warming levels of more than 6K are reached only in TDLR because here we prescribe a stronger homogeneous upper-tropospheric warming than in TD. This upper-tropospheric warming affects the surface warming trough adiabatic descent in the Mediterranean. We suggest that only the combination of the strong large-scale upper-level warming, and vertically near-adiabatic re-distribution of the warming evokes the Mediterranean amplification.

**Minor Comments:**

*Suggestion for lines 30-31: Since the dry adiabatic lapse rate is independent of temperature changes, the lapse rate changes driven by global warming are driven by the response of the moist-adiabatic lapse rate that decreases with warming.*

We included the nice sentence as suggested, and we think this improves the clarity.

*Line 44: In summer, however, also*

Corrected

*Lines 53-54: Do you really mean this? I think the argument is that lapse-rate changes accentuate the Mediterranean amplification, rather than cause it outright.*

You are right, this was too strongly formulated and might be misleading in the context of climate change. As stated in the title we mean that lapse-rate changes "regulate" the warming. We changed the wording to "govern" here.

*Line 71: are covering*

Thank you, this is corrected.

*Fig. 3c, e: Please remove the experimental mean, rather than the mean from the FCC experiment. This is a particular problem with panel e as I've noted above.*

We acknowledge the point and were usure how to plot it in the first draft as well. As the referee suggests we now use each simulation's domain mean to visualize the anomaly. We also added the absolute warming to the plot which we hope further improves the clarity.

*Line 134-135: The Mediterranean amplification is not absent in the TD experiment, it's merely reduced and it's hard to tell because the relevant mean temperature has not been removed.*

This was indeed an incorrect formulation and noted by both referees. As stated above we changed the figure to accord with the lower mean warming as in TD. We now state:

In contrast, the Mediterranean amplification is clearly weaker in TD where we prescribe no large-scale lapse-rate changes (Fig. 3g-i).

*Line 137-139: This is part of my major comment above and the place for a deeper analysis of the circulation differences between the TD and TDLR experiments. How does this dynamically weakened lapse rate accomplished? I think this is crucial for the argument.*

Thank you for pointing that out, we agree that this is an interesting point to further investigate. We have added an analysis of the vertical motions in the experiments (Figure 5) and overall found that the dynamic differences between TD and TDLR are small. Based on theory and previous literature we argue that the dynamically weakened lapse-rate is accomplished by vertical mixing of relatively dry air (especially adiabatic descent), which is now described in much more detail (see lines 166-217 in the revised manuscript).

*Line 146-148: I'm not sure I understand this sentence. We expect the upper tropospheric warming to be larger than surface warming no matter what due to climate change. I think clarification here would help me understand the argument.*

We agree that this was not well explained. We now state more clearly that lapse-rates within simulation domain change also in TD compared to what we impose at the lateral boundaries. The section now reads:

This follows from similar dynamic alterations of the imposed warming profiles as in TDLR, meaning that also in TD the lapse-rates adjust within the simulation domain compared to the warming profile imposed at the lateral boundaries. Also in TD, simulated lapse-rate changes are larger in Northern Europe than the Mediterranean (Fig. 3i). This leads to a stronger surface warming in the Mediterranean in agreement with TDLR (Fig. 3g,h), but the absolute magnitude of the warming is over 1 K smaller in TD.

*Fig 4: Maybe mask the oceans, the enthalpy changes there are so high it's a bit distracting from the argument you're articulating*

Good idea! We modified the figure as suggested.

*Lines 216-217: This seems at odds with the contention at other points in the paper (Lines 53-54) that lapse-rate changes cause the warming pattern.*

This is true, as stated above we changed the earlier statement and therefore left these lines unchanged.

*Lines 250-251: Again, I don't think the analysis as currently constituted shows that lapse-rate feedbacks are "decisive". Some numerical comparisons between the TD and TDLR warming amplification patterns over Europe could help here.*

We think that this conclusion is now backed with more data and more thoroughly shown and explained in the revised manuscript. Still, we weakened the wording here a bit and say that lapse-rate changes "regulate" the warming pattern, consistent with the title.

[revised manuscript text omitted]

The Mediterranean amplification is well reproduced by TDLR (Fig. 3d-f), the idealized simulations that are forced with a large-scale tropospheric lapse-rate change. In contrast, the Mediterranean amplification is  clearly weaker in TD where we prescribe no large-scale lapse-rate changes (Fig. 3 g-i). Quantitatively, the warming contrast between the Mediterranean

140 and Northern Europe is around 1 K weaker in TD than TDLR. The absolute warming in the Mediterranean is ∼5.6 K in TDLR, ∼4.5 K in TD, while it is ∼5.9 K in FCC.

To understand the reason for the difference between the two idealized simulations, we compare the respective vertical warming profiles (Fig. 3f,i). By design, the shape of the initial warming profile imposed at the lateral boundaries (shown by the gray lines in Fig. 3f,i) differs between the two simulations. In TDLR (Fig. 3d), the lapse-rate change in the Mediterranean

145 dynamically weakened compared to the imposed profile and the surface warming increased as a consequence (Fig. 3f, orange vs. gray line). In Northern Europe, we see the opposite, meaning that the lapse-rate change simulated in the domain interior is stronger than what was prescribed, and the surface warming has decreased in response (Fig. 3f, blue vs. gray line). In other words, for the Mediterranean, some of the warming imposed at the lateral boundaries has been dynamically re-distributed from the upper troposphere to the lower troposphere, and vice-versa for Northern Europe.

150 In the TD experiments, where we impose vertically uniform warming at the lateral boundaries, the surface warming in both the Mediterranean and Northern Europe is lower than what was imposed at the lateral boundaries (Fig. 3i) (Lenderink et al., 2019). This follows from similar dynamic alterations of the imposed warming profiles as in TDLR, meaning that also in TD the lapse rates adjust within the simulation domain compared to the warming profile imposed at the lateral boundaries. Also in TD, simulated lapse-rate changes are larger in Northern Europe than the Mediterranean (Fig. 3i). This leads to a stronger

155 surface warming in the Mediterranean in agreement with TDLR (Fig. 3g,h), but the absolute magnitude of the warming is over 1 K smaller in TD. An equally strong regional maximum in Mediterranean surface warming, as in TDLR or FCC, doesn't appear in the absence of  a strong upper tropospheric warming maximum prescribed at the model boundary, even though we impose a comparatively high surface warming of ∼5.3 K (Fig. 3i) and the land-surface feedbacks in the model are fully interactive.

160 Summarizing Fig. 3, we find that two decisive changes in the climate system are needed to simulate the Mediterranean amplification. First, a strong and horizontally uniform large-scale upper tropospheric warming must be present. In our simulation this corresponds to the high warming (>7 K) at altitudes above 8 km which is imposed in TDLR but not TD. Second, the lapse-rate changes are dynamically altered during the simulation depending on the  European region and directly connected to the extent of summer surface warming.

[Figure]

**Figure 4.** Mean summer relative humidity in the historical CTRL simulation and changes in the idealized simulations. (a) Climatological relative humidity in CTRL for the average summer in the 1971-2000 period. (b) Change in relative humidity in the idealized TD simulation, representing RCP8.5 and the 2070-2099 period. (c) Same as (b) but for TDLR. (d) Same as (b) but for the FCC, which is the GCM-driven transient regional climate simulation.

165       Since the Mediterranean is dryer in summer than Northern Europe (Fig. 4), the dynamic alteration of the lapse-rate change profile supports the idea that moisture availability controls the strength of the lapse-rate changes in mid-latitudes during summer (Joshi et al., 2008; Fasullo, 2010; Byrne and O'Gorman, 2013a, b, 2018; Brogli et al., 2019a). The simulated local adjustment of lapse-rates across the troposphere suggests that the European summer atmosphere is vertically mixed by local convection and subsidence. Lapse-rate changes are thus connected to the decrease of the moist-adiabatic lapse

170 rate at warmer temperatures and radiative-convective equilibrium (Held and Soden, 2000). Small lapse-rate changes may occur where moist-adiabatic vertical mixing is inhibited by dry conditions and temperature-independent dry-adiabatic mixing plays a larger role. This is likely the case in the Mediterranean summer season, where the relative humidity is around 40 % (Fig. 4a). Moist adiabatic vertical motions are infrequent (due to low moisture availability) and thus vertical warming gradients are small (there would be no vertical warming gradient if vertical mixing of warming throughout the troposphere was

175 entirely dry adiabatic). In Northern Europe, the summer season relative humidity is around 80 % (Fig. 4a), and is projected to increase further (Fig. 4b-d). Thus, vertical mixing in this region is more likely to follow a moist adiabat, which acts to change the lapse rates. The resulting substantial vertical warming gradients then lower the surface warming relative to the Mediterranean, according to the simulations shown in Fig. 3. Note that both regional climatological differences in relative humidity as visible in Fig. 4a, as well as relative humidity changes in response to warming (Fig. 4b-d) can contribute to the

180 differences in lapse-rate changes projected by simulations. Yet, in both TD and TDLR we observe quite moderate changes in relative humidity compared to FCC (Fig. 4b-c vs. d). Therefore, it is likely that in our idealized simulations, the climatological spatial differences in moisture availability are crucial to induce changes in lapse rates. Despite the presented evidence, from our simulations alone, we are unable to diagnose that humidity differences are the ultimate cause of the different lapse-rate changes simulated within the model domain. Yet, this connection has been shown theoretically and in idealized simulations in

185 earlier studies (Joshi et al., 2008; Byrne and O'Gorman, 2013a, 2018; Buzan and Huber, 2020).

      Lapse-rate changes affect the vertical stratification and thereby vertical motions in the atmosphere, which is further explored in Fig. 5. From the previous findings in Fig. 3f,i we observed that the tropospheric warming imposed at the lateral boundaries is vertically re-distributed within the simulation domain. A remaining question is if there are important dynamic adjustments

190 in the idealized simulations that control the vertical exchange (e.g. increased subsidence). Figure 5 shows the vertical wind ($w$) in the historical CTRL simulation along with the changes in TD, TDLR, and FCC for the simulations based on HadGEM2-ES. In the historical climatological summer mean our simulations show subsidence in the southern part of the domain and no mean vertical wind in the northern part of the domain (Fig. 5a), which suggests that in the north upward and downward winds cancel on climatological timescales as is characteristic for the extratropics. Neither TD nor TDLR show substantial changes in vertical

195 wind (Fig. 5b-c). In contrast, the subsidence slightly weakens in FCC (Figure 5d). Thus, in TD and TDLR surface warming contrasts develop without substantial dynamic changes (Figs. 5 and A1). Physically we can interpret the extra warming in the southern part of the domain using the thermodynamic equation. When written using potential temperature $\theta$, this is

$$\frac{D\theta}{Dt} = \frac{\partial\theta}{\partial t} + u\frac{\partial\theta}{\partial x} + v\frac{\partial\theta}{\partial y} + w\frac{\partial\theta}{\partial z} = \dot{\theta}, \tag{1}$$

[Figure]

**Figure 5.** Meridional cross-section of mean summer vertical wind (positive values mean upward motion) in the historical CTRL simulation and changes in the idealized simulations. (a) Vertical velocity in CTRL for the average summer in the 1971-2000 period. (b) Change in vertical in the idealized TD simulation, representing RCP8.5 and the 2070-2099 period. (c) Same as (b) but for TDLR. (d) Same as (b) but for the FCC, which is the GCM-driven transient regional climate simulation. This figure shows only data from the simulations based on the GCM HadGEM2-ES since the necessary three-dimensional model output was only stored for these simulations.

where $(u, v, w)$ denotes the three-dimensional wind vector, and $\dot{\theta}$ the diabatic heating rate. Equation 1 is applied to the slowly evolving mean flow, and $\dot{\theta}$ will thus include eddy contributions. Consider now the effect of the term $w(\partial\theta/\partial z)$. For the sake of the argument, we assume it is the dominating term, i.e. $\partial\theta/\partial t \approx -w(\partial\theta/\partial z)$. In the simulations TD and TDLR $w$ remains almost constant (Figure 5b-c). However, the increased stratification $\partial\theta/\partial z$ implies that the contribution of $w(\partial\theta/\partial z)$ to the local warming increases as well. In essence, the same subsidence with in an enhanced stratification implies an increased warming. The argument illustrates that the extra warming in the southern part of the domain results essentially from adiabatic descent. Since we only change $\partial\theta/\partial z$ at the lateral boundaries in the TDLR simulation this argument is especially relevant in TDLR.

Bringing together the results from Figs. 3, 4, and 5, we interpret the results as follows: Warming which we artificially impose in the idealized simulations is vertically re-distributed throughout the troposphere. The vertical motions required to achieve the re-distribution of the warming are already present during summer in the CTRL simulation (Fig. 5). The warming and moistening of the atmosphere in the TD simulation suffices to induce lapse-rate changes. These lapse-rate changes are more pronounced in present-day moist regions than in dry regions (Fig. 3i and Fig. 4a) because the vertical mixing follows the temperature-dependent moist-adiabatic lapse rate more frequently in moist regions. While in both TD and TDLR the lapse-rates locally adjust according to the near-surface humidity and in both simulations lead to an amplification of the Mediterranean warming, the very high warming levels of more than 6K are reached only in TDLR because here we prescribe a stronger homogeneous upper-tropospheric warming than in TD. This upper-tropospheric warming affects the surface warming trough adiabatic descent in the Mediterranean. We suggest that only the combination of the strong large-scale upper-level warming, and vertically near-adiabatic re-distribution of the warming, evokes the Mediterranean amplification.

Overall, during European summer, the entire troposphere is relevant to understand the surface warming pattern: From TD, we can observe that surface moisture gradients influence the warming up to 10-12 km height (Figure 3i). On the other hand, the extra upper tropospheric warming introduced in TDLR strongly affects near-surface warming levels (Figure 3f).

As discussed, the homogeneous and strong upper tropospheric warming is a key process in understanding Southern European summer warming. Yet, it may be surprising that the middle to upper tropospheric temperature change in summer over Europe is spatially almost uniform. While this seems clear in the simulations we analyzed, the physical reasons behind it remain more speculative and can be an avenue for future research. Generally, a strong upper tropospheric summer warming can be expected, since most of the globe is covered by oceans where one would expect strong lapse-rate changes due to the abundance of moisture. The uniformity of the warming could be related to the weak equator-to-pole temperature gradient in summer, which also results in weak baroclinicity, implying that warming gradients will also be small. Also, the long 30-yr periods that we averaged to obtain the results might act to smooth upper atmospheric warming gradients generally.

**3.3 Changes in moist enthalpy in EURO-CORDEX**

[Figure]

**Figure 6.** Decomposition of the anomaly of surface moist enthalpy ($H$) changes in the 0.11° EURO-CORDEX ensemble for the summer season over land. (a) Overall anomaly (deviation from the domain mean) of the change in moist enthalpy, normalized by the specific heat of air ($c_p$) to yield the unit Kelvin. Thus, the surface moist enthalpy change has been computed as $\Delta H = \Delta T + L_v\Delta q/c_p$. The anomalies of the two components of $H$ are shown separately. (b) shows the anomaly of $\Delta T$, while (c) shows the anomaly of $L_v\Delta q/c_p$. Gray shading shows ocean areas which are masked to improve the clarity.

235    Previously, we argued that the differential importance of moist- and dry-adiabatic vertical mixing controls the magnitude of summer lapse-rate changes and regulates the surface warming. A similar argument can be made when analyzing the change in moist enthalpy (Berg et al., 2016; Byrne and O'Gorman, 2018; Matthews, 2018). The moist enthalpy is given by $H = c_pT + L_vq$, where $c_p$ is the specific heat of air, $T$ is temperature, $L_v$ the latent heat of vaporization, and $q$ the specific humidity. In a warming climate, the change in moist enthalpy quantifies the combined effect of changes in internal energy given by

240    temperature changes ($c_p\Delta T$) and the change in latent energy, which could be released by condensation if air was lifted to the top of the atmosphere ($L_v\Delta q$).

Figure 6 shows the spatial anomalies of the overall change in moist enthalpy ($\Delta H$) and the two components $c_p\Delta T$ and $L_v\Delta q$ seperately. The data is from the EURO-CORDEX ensemble and the summer season (JJA). Note that over land, the simulations project a spatially relatively homogeneous change in $H$ (Fig. 6a). This means that total amount energy used for

245    raising temperatures or humidity is similar troughout the European continent. Yet, the change in internal energy or temperature shows a regional maximum the Mediterranean (Fig. 6b) while the change in latent energy is smaller in the Mediterranean than

elsewhere in the domain (Fig. 6c). The change in latent energy describes the potential for latent energy release by convection, which is also the root of lapse-rate changes. Thus, the change in moist enthalpy supports the notion that the additional available energy connected to climate change in the Mediterranean rather translates to an increase in surface temperature ($c_p \Delta T$) than to an increase in convective latent heat release ($L_v \Delta q$) and vice-versa for Northern Europe. The below-average increase in $\Delta q$ (Fig. 6c) is a clear sign for limited moisture availability in the Mediterranean, because from the climatological temperatures alone, one would expect a above-average increase of $\Delta q$ in the relatively warm Mediterranean (the warmer air could potentially carry more water vapour). This below-average increase in $q$ also implies a decrease in near-surface relative humidity, which means that the limited present-day Mediterranean water availability (Cramer et al., 2018), will intensify. Thus, both the present-day dryness (Fig. 4) and a low increase in atmospheric moisture limit the increase in latent energy in the Mediterranean.

[revised manuscript text omitted]